# Coaching and Prompting for Remote Physical Activity Promotion: Study Protocol of a Three-Arm Randomized Controlled Trial (Movingcall)

**DOI:** 10.3390/ijerph16030331

**Published:** 2019-01-25

**Authors:** Xenia Fischer, Lars Donath, Kimberly Zwygart, Markus Gerber, Oliver Faude, Lukas Zahner

**Affiliations:** 1Department of Sport, Exercise and Health, University of Basel, 4052 Basel, Switzerland; k.zwygart@stud.unibas.ch (K.Z.); markus.gerber@unibas.ch (M.G.); oliver.faude@unibas.ch (O.F.); lukas.zahner@unibas.ch (L.Z.); 2Department of Intervention Research in Exercise Training, German Sport University Cologne, 50933 Köln, Germany; l.donath@dshs-koeln.de

**Keywords:** physical activity promotion, remote, telephone coaching, SMS prompting, inactive adults

## Abstract

*Background*. Physical inactivity is currently seen as one of the biggest global public health issue. Remote physical activity (PA) promotion programs are expected to be effective if they are individually tailored and include behavior change techniques, personal coaching, and regular prompting. However, it is still not fully understood which intervention components are most effective. This paper describes the rationale and design of a study on an individually tailored remote PA promotion program comparing the efficacy of coaching and prompting with a single written advice. *Methods*. In total, 288 adults (age 20 to 65 years) were randomly assigned to three different intervention arms of a 6-month-long PA promotion program. A minimal intervention group received a single written PA recommendation. The two remaining groups either received telephone coaching sessions (*n* = 12 calls) with or without additional short message service (SMS) prompting (*n* = 48 SMSs for each participant). Data assessment took place at baseline, at the end of the intervention, and after a six-month follow-up-period. The primary outcome of the study was self-reported PA. Objectively assessed PA, psychosocial determinants of PA, well-being, body mass index (BMI), and adherence were assessed as secondary outcomes. *Conclusion*. Findings of this three-arm study will provide insight into the short and long-term effects of coaching and prompting for PA promotion.

## 1. Introduction

Physical inactivity is associated with various non-communicable diseases and rising health care costs. As an independent risk factor for cardiovascular disease, physical inactivity accounts for 9% of premature deaths globally [1]. Adults should engage in at least 150 minutes of moderate or 75 minutes of vigorous physical activity (PA) per week [2]. Based on self-reports, approximately 30% of the adult population worldwide as well as in Switzerland do not reach these minimal recommendations [3,4]. Physically inactive lifestyles have been linked to technological developments, mechanization, an increased use of motorized transport, and an increase in sedentary leisure-time activities [4].

The lack of time and motivation represent the main self-reported barriers for leisure time PA in working-aged adults [3,5]. To achieve PA-related behavior changes, adults need support to overcome personal barriers.

Environmental approaches, interventions in social settings (e.g., the workplace), and personal interventions that focus on motivational aspects have been proven to be effective in promoting PA [6]. On an individual level, multicomponent strategies are used to encourage participants to incorporate regular PA in their daily lives. Individually tailored interventions (which consider demographic characteristics and the current behavior) based on empirically validated theories are considered most effective [7]. 

During recent years, an increasing interest has been observed in interventions delivered without a face-to-face interaction. Mobile phone or web-based communication modes are commonly available and enable reach to a wide population. [8]. Compared to face-to-face interventions, remotely delivered programs are considered more cost-effective [9]. An additional benefit is the reduction of time, transportation, and personal contact barriers (i.e., fear of prejudice) [10]. Within remote interventions, telephone contact to provide feedback or to facilitate behavior changes were most effective for promoting self-reported PA [8]. The personal communication enables a further individualization of intervention content and the use of health-coaching techniques. According to Olsen [11], health-coaching refers to a goal-oriented, client-centered, and health-focused interactive partnership between the coach and the participant that is based on a process of enlightenment and empowerment. Combined with other modalities (e.g., print), telephone delivery has shown similar effects to face-to-face interventions [12]. Hence, telephone-based coaching lasting 6–12 months that includes 12 or more calls has been proven effective [13]. The potential to disseminate individually tailored telephone coaching has recently been demonstrated in a population-based study in Australia. After a six-month period with 10 coaching sessions, 1440 participants reported significant improvements in PA levels and body weight [14]. 

Web-based behavior change interventions include the use of (self-navigated) educational information, computer-tailored feedback, goal-setting and self-monitoring applications, and/or access to a counsellor via e-mail or through chat rooms [15]. An existing meta-analysis indicates positive changes in PA levels [16]; however, effect sizes are small. Depending on the level of interactivity, web-based interventions suffer from low engagement and high retention that are associated with decreased effectiveness [16]. 

Periodic prompts represent another remote communication strategy for PA promotion [17]. Periodic prompts are messages sent multiple times without a user-initiated request. They are often delivered as short message services (SMSs) by phone. These intervention have the advantage that 90% of obtained SMS are read within minutes once received [18]. Messages are used as a reminder, to provide feedback, or to share information on strategies to facilitate behavior change [17]. A previous meta-analysis showed a small-to-moderate positive effect of SMS prompting on preventive health behaviors [19]. Thereby messages sent at varying times though out the day and the week seem to be more effective, as they prevent habituation and communicate more “social presence” [20].

Behavior change techniques (BCTs) constitute the “active ingredients” of PA interventions. Independent of their delivery mode and context, BCTs can be categorized according to the V1 Taxonomy developed by Michie et al. [21]. Previous meta-analyses pointed out the effectiveness of certain BCTs to change PA behavior [22,23,24]. In particular, *self-monitoring of behavior* has been associated with intervention effectiveness [24]. Self-monitoring requires the recording of frequency, intensity, or type of PA and makes individuals regularly aware of their current behavior [25]. Existing meta-analyses and reviews [22,23,24,26,27,28,29,30,31] further underline the importance of *goal setting* concerning the behavior and the outcome of the behavior. They additionally emphasize the use of *action planning*, *social support*, *problem solving*, *feedback on performance*, *review of behavior goals*, *instruction on how to perform the behavior*, and *information about health consequences*. Furthermore, a recent meta-analysis by Samdal et al. [32] showed the importance of an autonomy-supportive and person-centered communication mode.

Increases in PA are accompanied by improvements in overall well-being [33] and changes in psychological variables which moderate PA participation (e.g., self-efficacy) [31]. Only few studies examined the effects of PA promotion after a follow-up period without intervention and thus data on long-term outcomes are scarce [8,34]. Additionally, PA levels are usually assessed via questionnaires (e.g., 7-Day Physical Activity Recall [35], International Physical Activity Questionnaire [36]) which are effected by recall and social desirability bias. To date, objective measures (e.g., accelerometer, pedometer) have rarely been used to examine the impact of interventions to promote PA [37].

In summary, there is clear evidence for short-term behavior changes through remote delivery of BCTs. However, it remains unclear which intervention components are most effective and can be translated into practice. In order to gain a better understanding of which specific elements of an intervention result in behavior change, studies with multiple intervention arms and different combinations of intervention components are needed [38]. Furthermore, the maintenance of PA-related behavior change needs to be evaluated across longer periods of time.

Aim of the Study

The main goal of the “Movingcall” study is to evaluate a remote individually tailored PA promotion program based on individually assigned BCTs. More specifically, the following three research questions are addressed: First, what is the short-term effect of remote telephone coaching with and without SMS prompting on self-reported and objectively assessed PA levels in inactive adults compared to a control condition (written information about recommended levels of PA; minimal intervention)? Second, what is the long-term effect (six-month follow-up without contact with participants) of the intervention compared to a control condition? Third, how does the intervention impact on psychological determinants which are involved in the regulation of exercise behavior, and do these determinants mediate the effect of the intervention on PA behavior?

## 2. Materials and Methods 

### 2.1. Study Design

Movingcall is conceptualized as a three-arm randomized controlled trial (Figure 1). The three study arms differ in terms of the delivery mode of the intervention: A “control group” received a single written recommendation at the beginning of the intervention (minimal intervention). A “coaching group” received 12 biweekly telephone coaching sessions. A “coaching and SMS group”, similarly received 12 biweekly telephone coaching sessions which were extended by two SMS messages per week. Data assessment took place at baseline, at the end of the six-month intervention, and after a six-month follow-up period without contact with the participants. The study was conducted at the Department of Sport, Exercise, and Health at the University of Basel, Switzerland.

### 2.2. Recruitment and Sample Size 

Participants were recruited through newspaper advertisements, e-mail newsletters, advertisements provided by local and cantonal administration offices, companies’ communications to employees, and flyers in pharmacies, medical practices, and physiotherapy clinics, as well as by word-of-mouth publicity. Individuals who were interested in participating contacted the study team by e-mail or by phone. Eligible individuals were provided with a written informed consent before participating in the study.

Based on previous studies on remote PA promotion and the comparison of SMS prompts with alternative interventions [8,20] a small effect (d = 0.21) on self-reported PA was expected. Based on this expectation, a sample size calculation for an analysis of covariance (ANCOVA) including baseline PA as a covariate was computed [39]. In order to achieve a power of at least 80% (1-β error probability) and a significance level of 5%, a total of 242 participants were required. A subsequent total sample size of approximately 284 participants was needed after adjusting for an expected dropout rate of 15–20%.

### 2.3. Inclusion Criteria 

Men and women between 20 and 65 years of age, who failed to meet the recommended 150 minutes of moderate PA [2] per week, were eligible for the study. Additionally, individuals had to understand German sufficiently in order to complete the study procedures. They also had to have residence in Switzerland. Inclusion criteria were checked using a questionnaire sent by e-mail. Physical activities of the previous week were assessed by an adapted version of the Simple Physical Activity Questionnaire (SIMPAQ [40]). Participants were further screened using the Physical Activity Readiness Questionnaire (PAR-Q [41]). Individuals who reported one or more health concerns in the PAR-Q were asked to consult their general practitioner (GP). If their GP approved moderate PA, individuals were deemed eligible. Individuals who reported more than 150 minutes of moderate PA or who shared the same household with another study participant were excluded. Further exclusion criteria were current pregnancy or a planned absence of more than 3 weeks during the next 6 months. 

### 2.4. Group Allocation

A computer-based minimization procedure [42] stratified by age and sex was used to assign participants randomly to one of the three study arms. The randomization was conducted by a member of the research team, who was not involved in the intervention. In order to simplify the allocation of participants within the study team, randomization occurred prior to baseline data collection. Neither the study team nor the participant knew the group allocation prior to the start of the intervention.

### 2.5. Program Development

#### 2.5.1. Theoretical Foundation

The intervention content (e.g., BCTs) and delivery modes (e.g., frequency and density of participant’s contact) were defined according to the evidence of previous studies in the area of behavior change research and to theoretical considerations. 

The Behavior Change Wheel (BCW) framework [43,44] serves as the theoretical background of the intervention. The BCW allows a holistic consideration of behavior change interventions and summarizes existing behavior change theories. It clarifies the intervention functions (e.g., incentivization, training), policy levels (e.g., regulation, service provision), and BCTs which can influence behaviors. The theoretical domains framework of the BCW summarizes psychosocial determinants (e.g., social influences, beliefs about capabilities, goals) that explain the occurrence of a behavior in the so-called theoretical domains [45]. Each theoretical domain is influenced by certain BCTs [45]. Consequently, the selection of BCTs can be conducted theoretically by considering the theoretical domains. In contrast to various behavior theories, the BCW framework does not explain causal relations between the theoretical domains. Therefore, the MoVo (acronym for motivation and volition) Process Model by Fuchs et al. [46] serves as the theoretical basis to explain PA specific causal relations of theoretical domains. The MoVo Process Model combines concepts of stage theories (motivational and volitional phases of behavior change) and continuous behavior modes (causal relations). It represents an adaption of the Health Action Process Approach (HAPA) by Schwarzer [47] for PA-related behavior change. Psychosocial determinants of the MoVo Process Model are assessed as secondary outcomes (see Section 2.7.3) and used to tailor the intervention content individually (see Section 2.6.1).

#### 2.5.2. Intervention Content

The Movingcall program communicates BCTs and knowledge based on exercise science. The goal is to increase PA gradually and to reach the recommended 150 minutes of moderate PA per week or more [2]. Ten BCTs that were proven to be effective in changing PA behavior according to previous meta-analyses were selected (see background) [22,23,24,26,27,28,29,30,31]. For the present selection of BCTs, meta-analyses that defined BCTs according to an older taxonomy [48,49] were translated to the current V1 taxonomy. The resulting ten BCTs are described in Table 1. These BCTs were delivered to all participants.

The Movingcall program allows for 25 additional BCTs (listed Table 2). This includes BCTs that might be implemented unintentionally within the conditions of a regular coaching session (e.g., set graded tasks, focus on past success) and BCTs with scarcer evidence of effectiveness in previous studies (e.g., prompts/cues, time management, social and self-reward [23,27]). Those BCTs are delivered according to the need of the participant. The inclusion of additional BCTs should ensure that all delivered BCTs are identified and documented.

All the BCT definitions and practical examples for PA promotion are gathered in an online manual. The online manual includes information on the optimal timing to deliver the BCT as well as information on what theoretical domains are mainly influenced by the BCT [50,51]. BCTs are explained according to the V1 taxonomy by Michie et al. [21]. To simplify documentation, the BCTs subcategories of *social support* (i.e., *unspecified*, *practical*, and *emotional*) are merged. Similarly, two BCTs of *natural consequences* (*salience of consequences* and *information about emotional consequences*) and four BCTs of *reward and threat (social reward, social incentive, self-incentive*, and *self-reward)* are merged. The BCTs *time management* and *provide information on where and when to perform* are defined according to CALO-RE taxonomy [48].

The Movingcall program further comprises exercise science based advice for previously inactive adults. This includes information on a reasonable initial regime and increasing endurance and resistance training with regard to frequency, duration and intensity (e.g., start with 30 minutes brisk walking and increase to 45 minutes within the next three weeks). Further topics are the inclusion of PA in everyday life (e.g., plan walking distances, go to work by bike, balance training in the office), advice regarding balance training as well as stretching, and recommendations regarding movement or sports (e.g., interval training for Nordic walking). This information is gathered in our written guidelines.

### 2.6. Intervention Delivery

#### 2.6.1. Tailoring

Intervention content of all study arms is tailored individually to participants’ preferences and needs. This includes tailoring on demographics (e.g., age, place of residence), on the current behavior (e.g., form of activities, amount of PA per week), and on theoretical domains that influence the behavior (e.g., social support). Furthermore, information about participants’ intention, planning, pros and cons, self-efficacy, barriers, coping strategies, social support, knowledge on PA recommendation, PA behavior, personal goals and constraints, and current living conditions are used for tailoring. The required information is retrieved during the pre-test period. A focus is placed on BCTs that influence theoretical domains with the most potential for development. Depending on the study arm, the tailoring will be continued dynamically during the coaching process.

#### 2.6.2. Activity Profile

Participants in all study arms (including the control group) have access to their personal activity-profile on an interactive homepage (www.movingcall.com; Figure 2) [52]. The platform is designed to enable and standardize the delivery of the BCTs *action planning* and *self-monitoring*. The password-protected profiles can be used as a stand-alone tool or can simplify the interaction between participant and coach. To actualize the coaching process, each coach has access to the profile of the corresponding participant. Within the profile physical activities are scheduled and documented on a “plan page”. Activities can be selected from a database and entered into the participants’ plan. In order to self-monitor the PA behavior, completed activities need to be checked and additional activities should be entered. Participants are encouraged to document all activities lasting 10 minutes or more. The homepage provides kilocalories (kcal) and the metabolic equivalent (MET) of entered PA according to the compendium [53]. The profile further includes a note board to simplify communication between coach and participant. Interaction with other participants is not possible. Lastly, the homepage includes an online questionnaire tool. Participants use their login to answer the questionnaires and the coaches can access to the answers. The use of the profile is explained to all participants by phone prior to the start of the intervention. Participants are given access to their profiles during the intervention and during the follow-up period.

#### 2.6.3. Telephone Coaching (Coaching Group)

Participants in the coaching group receive twelve biweekly telephone coaching sessions. Coaching sessions last 15–20 minutes during the course of the intervention and 30–40 minutes in the first two sessions. Participants are called by the same coach for the entire intervention period. If the biweekly rhythm is interrupted (e.g., due to vacations) or if participants miss a call, coaches can reschedule the session. Occasional variation of the biweekly rhythm between 1 and 3 weeks between each coaching session is considered normal. After each call the coaching session is summarized in a few sentences on the “note board” of the participant. Participants and coaches can leave remarks on the note board but it does not serve as a chat function.

The coaching refers to an interactive discussion between participant and coach. In accordance with the definition of health coaching by Olsen [11], the sessions are goal-oriented, client-centered, and focused on PA. Questions on nutrition or other health-related concerns as well as unrelated topics are not answered in detail. Coaches encourage their participants to build autonomy, self-confidence and self-efficacy and to gain knowledge and experiences concerning the benefits of PA. 

All coaching sessions contain the elaboration of BCTs and advice on training concepts. The scope of each specific coaching session is stated in the written guidelines. During the first sessions, coaches focus on the development of a functional relationship and deliver information on the procedure of the coaching. Additionally, coaches start to encourage participants to set goals and to plan their physical activities. Established activity plans are inserted in the personal profile on the homepage and visible for participant and coach. With support of the coach, participants proceed to the following session by establishing coping strategies to overcome personal barriers. All main BCTs and additional BCTs are delivered as required in the 12 sessions. Based on the self-monitoring data which is visible on the profile and based on the discussion during the coaching sessions, participants receive feedback concerning their behavior change towards a physically active lifestyle. Towards the end of the 12 sessions, coaches focus on the maintenance of PA levels.

#### 2.6.4. Telephone Coaching and SMS Prompting (Coaching and SMS Group)

Participants allocated to the coaching and SMS group receive the same telephone coaching as the coaching group. Additionally, participants in this group receive four SMS prompts during each two-week period (48 in total). The SMS are written, sent, and timed by the corresponding coach through an online platform. SMSs have the consignor of Movingcall and it is not possible to respond. Participants are asked if they have a preferred time (morning, noon, afternoon, evening) to receive the prompts. Within this time the SMSs are sent at variable time points. During each two-week period participants receive one SMS relating to PA knowledge, one relating to discussed BCTs, one containing a reminder, and one that consists of a feedback on documented physical activities (examples in Table 3). The language style and length of the SMSs are standardized and all SMSs are collected and saved.

#### 2.6.5. Intervention Provider

The coaching sessions are held by 28 trained sport science and psychology students. The training of the coaches consists of ten lectures and practical exercise for telephone coaching. Coaches are provided with an online handbook on BCTs and written guidelines. The following topics are included in the training: (1) The application of BCTs and the theory of behavior change; (2) An autonomy-supportive and client-centered coaching style; (3) Knowledge based on exercise and movement science to advice previously inactive adults; (4) The tailoring procedure; (5) Standardization of the intervention delivery (including written recommendation and SMSs); and (6) Data collection and documentation of the intervention. Each coach takes part in a one-hour know-how-check prior to the first coaching session, to ensure the coaching skills are sufficient. During the intervention delivery, current challenges in coaching situations are discussed and above listed topics are deepened in biweekly team meetings. 

#### 2.6.6. Intervention Documentation and Assessment of Adherence

Applied BCTs are documented and coded by the coaches after each coaching session. In addition, the duration, date and time of each session is recorded. 

The adherence of participants to the intervention is assessed in two ways. First, the engagement of participants is assessed based on their personal profile. The amount of active edits of each participant within the personal profile is recorded (e.g., entering a planned PA). Second, the completeness of intervention delivery is assessed by documenting date, duration and number of attended coaching sessions. Thereby the subcategories “standard”, “not standard”, “non-usage attrition”, and “dropout” are distinguished [54]. A total of 12 coaching sessions with up to two interruptions of maximum four weeks between two coaching sessions and an overall intervention duration of 22 (norm) to 26 weeks are considered as standard. Participants who stop the coaching (non-usage attrition) are asked for possible reasons and asked to still participate in the post and follow-up-tests. Individuals lost to post and follow-up-tests are considered as dropouts. 

#### 2.6.7. Minimal Intervention (Control Group)

Participants allocated to the minimal intervention group receive a single written recommendation about health-enhancing PA at the beginning of the intervention. The recommendation is tailored individually and covers information on all main BCTs. An exemplary PA plan and an explanation on how to adapt the plan is included. Participants in this group are asked to follow the recommendation, to self-monitor and plan their PA within their profile on the homepage and by doing so increase PA gradually during the intervention period. Recommendations are written by the coaches within a predefined template. Participants receive their recommendation as a PDF by e-mail and it is inserted on the note board within the personal profile. Participants in the minimal intervention group only have contact with the study team during data assessments.

### 2.7. Assessment of Primary and Secondary Outcomes

The assessment of primary and secondary outcomes is conducted without on-site presence of participants. Self-reported and objectively assessed PA level of one week, theoretical domains of PA as well as health and wellbeing-related variables are assessed in the pre, post and follow-up tests. After 3 months of intervention all participants are asked to answer a feedback questionnaire. Personal data and socio-demographic information are assessed during the inclusion procedure. Measures of all test periods are displayed in Table 4.

#### 2.7.1. Self-Reported Physical Activity

A structured telephone interview is carried out to assess participants’ self-reported PA. This interview follows the questions described in the SIMPAQ (Simple Physical Activity Questionnaire) [40]. In the present study, self-reported PA serves as the primary outcome. The SIMPAQ refers to PA during the previous 7 days. Furthermore, the structured interview asks for the time spent in bed and the time spent with the activities “sitting or lying”, “standing”, “walking”, and “other activities” (e.g., house work, gardening) for one of the usual days of the previous week. The duration and intensity of athletic training and recreational physical activities are assessed separately for each day. SIMPAQ was validated with university students by Schilling et al. [70] and has shown a moderate to strong correlation to accelerometer data and to established self-report questionnaires (Seven-Day Physical Activity Recall [71] and the short form of the International Physical Activity Questionnaire [72]). Compared to other self-report questionnaires the SIMPAQ is less time consuming [70]. 

In the present study, the original version of the SIMPAQ interview is extended to assess moderate or vigorous activities in everyday life. The intensity of the time spent walking and the time spent with other activities is specified by the question “how much of this time did you walk/move with an intensity that leaves you slightly out of breath?” Additional questions are added to the interview to assess: (1) If the last week is considered “a normal week”; (2) Illnesses and injuries during the last week; (3) Illnesses and injuries during the past 6 months including days of occupational absences; (4) Regular medication; and (5) Body weight including the time point and the location of weighing. In the post and follow-up tests two more questions are added to assess: (6) Other health-related behavior change attempts (e.g., stop smoking); and (7) The attendance to other services that support PA-related behavior changes (e.g., physiotherapy).

Interviews are conducted by specifically trained members of the study team. They are never conducted by the coach of a participant and interviewers are unaware of the group allocation.

#### 2.7.2. ActiGraph Data

PA of 1 week is objectively assessed with the ActiGraph wGT3X-BT. The triaxial accelerometer records intensity and duration of accelerations and converts the signals to “activity counts” [55]. Participants receive the device by post at the beginning of the pre-, post-, and follow-up assessment period. All participants are requested to wear the ActiGraph on the non-dominant wrist for 7 to 10 consecutive days and nights. Hip-worn accelerometers are regarded as superior for predicting locomotion and estimating energy expenditure [73]. However, estimates of moderate and vigorous PA of wrist- and hip-worn accelerometers are comparable [74]. While hip-worn accelerometers are today’s standard, wrist-worn accelerometers have demonstrated higher wear-time compliance [74,75]. Wrist-worn assessment is therefore considered more feasible for the remote setting. 

Each participant is called by the allocated coach to explain how to handle the ActiGraph and the procedures of the test period. Participants are asked to wear the device until the SIMPAQ interview that is arranged after the requested wear time. The ActiGraph should be ignored and only removed when swimming or in a sauna. Non-wear periods are reported on a sheet enclosed in the mail. A prepaid envelope is attached for returning the ActiGraph along with the sheet.

#### 2.7.3. Psychosocial Determinants of PA and other Measures

Online questionnaires are used to assess additional secondary outcomes and information needed for individual tailoring. Questionnaires are programmed within the interactive homepage. A link to answer the questionnaires is sent to the participant by e-mail. 

Applied instruments have previously been used in empirical research in German speaking countries. Except for self-composed items, acceptable validity and reliability have been demonstrated in previous studies (references listed below).

The following measures are assessed in the pre- post- and follow-up test-periods.
**Intention**: Motivational readiness for PA-related behavior change is assessed according to the stages of change of the Transtheoretical Model [56,76]. Participants select one out of five answers to the question “Are you regularly physically active, this means at least 20 minutes on 3 days of the week?”, e.g., “No, but I intend to become more physically active in the next 6 months”. A single item to measure the strength of intention is added. Participants quantify their intention to be regularly physically active on a scale from zero (no intention) to five (very strong intention) [57,63].**Action planning** is measured by five items [61]. Participants are asked if they defined when, where, how, how often, and with whom they plan to exercise. For example “I have made a detailed plan regarding when to exercise”. Answers are given on a Likert-scale from one (not true) to four (completely true).**Outcome expectations** regarding regular PA are assessed with 16 items validated by Fuchs [59]. Participants are asked to rate their expectations towards specific outcomes on a Likert-scale ranging from one (not true) to four (completely true). For example “If I were physically active on a regular basis, I would lose weight”.**Self-efficacy** is assessed consistently with Fuchs [60] by three items. The confidence to begin, to maintain and to restart regular PA is measured on a six-point Likert-scale from one (No confidence) to four (100% confidence). For example, “I am confident that I could start with new physical activity”.**Perceived barriers:** Participants are presented 19 potential barriers (e.g., “being tired”) and asked to indicate how strong each one prevented PA on a Likert-scale from one (not at all) to four (very much) [46]. **Coping strategies** on barriers are measured by five items applied and validated by Sniehotta et al. [58]. For example “I have made a detailed plan regarding what to do if something interferes with my plans”. Responses are given on a four-point Likert-scale from one (not true) to four (completely true).**Social support** is assessed by seven items that rate the confidence for support of the social environment on a four-point Likert-scale from one (never) to four (always) [62]. For example, “I am confident that people of my social environment will be physically active with me”.**Self-concordance** is measured by four items on the self-concordance scale. The scale was composed and validated by Seelig and Fuchs [63]. Participants are asked to rate their internal/external motivation on a Likert-scale ranging from one (completely false) to four (completely true). For example, “If I am physically active within the next weeks and months, this is because other people say I should”.**Knowledge about health enhancing PA and fitness rating:** Participants are asked to rate their fitness level (on a scale from one to ten, one item) [64] and their level of health-enhancing PA (“Do you think you are sufficiently active for your health? Yes/No”, one item) [65]. Knowledge of general PA recommendations are assessed by two items previously used by Gerber et al. [66].**Perceived stress-related exhaustion symptoms** are measured by the validated Shirom–Melamed Burnout Measure (SMBM) [67]. Three subscales (physical fatigue, emotional exhaustion and cognitive weariness) are assessed within 14 items. **Health-related quality of life:** The Short–Form 12 Questionnaire (SF-12) is used to assess health-related quality of life [68,69]. The questionnaire includes 12 items on general physical health status and mental health distress. The questionnaire’s validity and reliability has been demonstrated by Craig et al. [36].

Two questionnaires are used to tailor the intervention content and are only applied in the pre-test. One questionnaire consists of family structure, living situation and workload in four self-compiled items. The other instrument assesses health restrictions, experience in PA and personal goals.

After 3 months each participant is invited to provide feedback on aspects of the intervention. The self-compiled questionnaire measures general satisfaction with the program, goal achievement, motivation to proceed and participant’s opinion on possible program extensions (e.g., interaction with other participants). Participants are asked to rate the usability of the personal profile, to evaluate the value of BCT-based advice and to name the most important recommendation for them personally. The control group is asked to rate the written recommendation. In the coaching and the coaching and SMS group the quality of the relationship between participant and coach is assessed. In the coaching and SMS group, it is evaluated whether the SMSs are perceived as beneficial and what topic is considered most motivating. Questions specific to group allocation are asked a second time in the post-test.

### 2.8. Data Processing and Statistical Analysis

ActiGraph data are analyzed using the ActiLife software (Version 6.13.3, ActiGraph, Pensacola, USA). Daily minutes of moderate-to-vigorous physical activity (MVPA) are computed after declaring sleep and non-wear time. Currently there are no widely used and validated cut-off-values to classify MVPA for adults’ activity count of wrist-worn ActiGraph data. The cut-off values are therefore chosen on the basis of existing explorative analysis [77] and their comparison to cut-off points for children [78]. According to Kamada et al. [77] activity counts per minute are classified as sedentary activity 0–1999, light activity 2000–8249, and MVPA > 8249. The option “worn on the wrist” provided by the ActiLife software is not selected given its systematic underestimation of energy expenditure [79]. Sleep is determined using the Cole–Kripke algorithm [80]. Non-wear time is calculated according to Troiano [81]. Physical activities listed on the non-wear sheet (e.g., swimming) are added as moderate activities. In order to be analyzed, participants need to have a non-wear time of less than 10% [82]. At least four weekdays and one weekend day are required [55]. The first day of wear-time is not included. For further analysis, the mean of MVPA per day of all assessed valid days is computed.

All statistical analysis will be conducted as intention-to-treat. Group differences in primary and secondary outcomes will be calculated using linear mixed models. Linear mixed models will include intervention group, time, group × time interaction, and baseline measures as fixed effects and subjects as random effects. Statistical analysis will be conducted using the statistical software STATA (StataCorp, College Station, USA). Effect sizes for pre-post change scores and the corresponding confidence intervals will be provided [83,84].

## 3. Discussion

The present paper describes the rationale, design, and content of a randomized controlled trial on remote PA promotion. The evaluation of the three study arms will enable conclusions on effects of coaching and prompting. Telephone coaching [12] based on suitable BCTs [24] as well as SMS prompting [17] has been proven effective to induce PA-related behavior changes. However, so far only few studies compared multiple intervention concepts to elicit, which concept should be translated into practice. Van Wier et al. [85] concluded that lifestyle counselling by phone as well as by e-mail is an effective tool to reduce body weight in comparison to a control condition. Regarding PA behavior changes, only telephone coaching showed significant improvements compared to the control group. Another study by Marcus et al. [86] and one by van Keulen et al. [87] detected no difference in PA levels comparing tailored print to telephone communication for PA promotion. To our knowledge, text messages in combination with telephone counselling sessions have only been conducted in weight loss and disease management studies [88] and have proven effective. Further studies, using multiple intervention arms to compare short- and long-term efficacy of specific intervention concepts are required [38]. Thereby, the provision of a detailed description of intervention contents will help identify evidence-based practice [12].

The present study will expand previous research and contribute to an increased understanding of remote intervention components for PA promotion in working aged adults. We hypothesize that coaching combined with prompting results in a higher and more sustainable increase in PA than coaching alone. We further hypothesize that both coaching groups outperform the minimal intervention group. The analysis of psychosocial determinants of PA will contribute to the theoretical explanation of PA behavior change. The effect of BCT based coaching on specific psychosocial determinants can be explored. Thereby, the precise documentation of applied BCTs will be valuable to further specify the content of the coaching. The three time points of measurements will further allow conclusions on psychosocial determinants relevant for maintenance of PA levels. Finally, the analysis of participants’ engagement with the program as well as their feedback will enable to analyze the feasibility of remote PA promotion in Switzerland.

The following strengths and limitations of the study should be considered. First, it is essential that participants do not visit a clinical institution during the entire study. In Switzerland, the main barriers for PA are “lack of time” because of occupational or other duties (40%), “lack of motivation” (18%), and “medical reasons” (18%) [89]. One of the advantages of remote PA promotion is the absence of travel time. In order to reach the target group of physically inactive adults, the remote setting needed to be preserved in the study setting.

The second strength/limitation is the objective assessment of PA. Studies on PA promotion commonly use self-reported measures to assess PA [8]. To overcome social desirability and recall bias, PA is additionally assessed by ActiGraph. Given the remote setting of the present study, it is considered essential that the ActiGraph is easy to handle and can be explained by phone. The decision for a wrist-worn assessment will probably lead to higher wear-time of the devices [75]. It is however a clear limitation that no validated cut-off points exist for wrist-worn ActiGraph data. Additionally, the possibility to compare the data with studies using hip-worn measures will be limited. Third, the present study implements a minimal credible intervention instead of a traditional waiting list or control group. This is considered important for two reasons: First, behavioral interventions cannot be blinded to the condition a participant is in. The minimal credible intervention (“placebo group”) is an acceptable possibility to overcome the bias of a control group [90,91]. Second, implemented intervention components have already proven effective in previous studies [8]. Therefore, the central objective is to compare their efficacy to a treatment that could be considered “treatment as usual”. Our minimal intervention is considered comparable with simple online tailored information or advice from a GP.

## 4. Conclusions

With regard to the increasing cost of non-communicable diseases, agencies in the health sector are currently looking for attractive personalized and affordable approaches in order to help individuals adapt a healthy lifestyle. Internet-delivered interventions, as well as coaching and prompting, are promising approaches to reach adults in Switzerland. The present study will contribute to the knowledge on how to improve PA levels effectively and sustainably in adults. Further insights into effects of coaching, prompting and tailoring strategies based on utilizing suitable BCTs on PA promotion can be delivered. 

**Trial registration:**ClinicalTrials.gov ID NCT02918578 registered on 23/09/2016.

**Ethics Approval and Consent to Participate:** The study protocol was approved by the ethics committee *Nordwest und Zentralschweiz*/EKNZ (ID: 2016-00560). The written consent of participants to participate was gathered prior to study inclusion.

## Figures and Tables

**Figure 1 ijerph-16-00331-f001:**
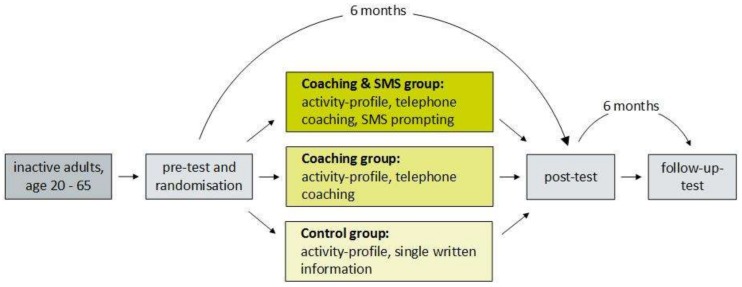
Study design. SMS: short message service.

**Figure 2 ijerph-16-00331-f002:**
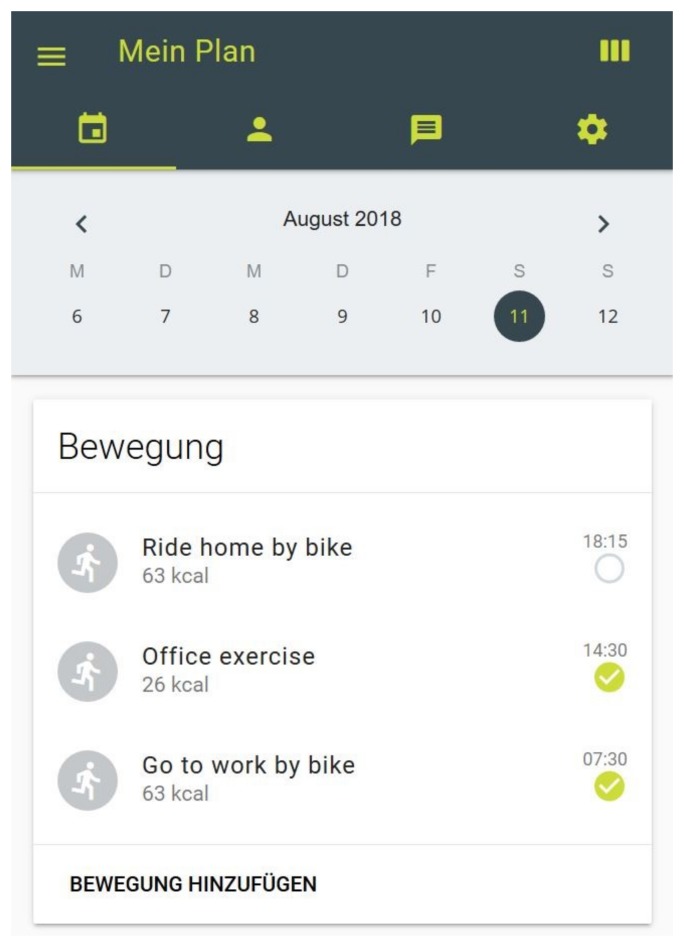
Plan page of activity profile on www.movingcall.com.

**Table 1 ijerph-16-00331-t001:** Applied behavior change techniques.

BCT (V1 Number)	Definition
Goal setting (behavior) (1.1)	Set or agree on (a) behavioral goal(s) that leads to increased PA.
Problem solving (1.2)	Analyze, or prompt the person to analyze factors influencing PA and generate or select strategies to overcome barriers and/or increase facilitators.
Action planning (1.4)	Prompt detailed planning of PA (must include at least one of context, frequency, duration and intensity). Context may be environmental (physical or social) or internal (physical, emotional or cognitive).
Review of behavioral goal(s) (1.5)	Review PA goal(s) jointly with the person and consider modifying goal(s). This may lead to re-setting the same goal, a small change in that goal, or setting a new goal instead of (or in addition to) the first, or no change.
Feedback on behavior (2.2)	Give informative or evaluative feedback on monitored (including self-monitored) PA.
Self-monitoring of behavior (2.3)	Prompt self-monitoring and recording of PA (i.e., diary).
Social support (3.1)	Advise on, arrange or provide social support (e.g., from friends, relatives, colleagues, buddies or staff). This may include practical help (3.2) and emotional support (3.3)
Instruction on how to perform the behavior (4.1)	Teach skills and knowledge required for specific physical activities, i.e., give clear instructions.
Information about health consequences (5.1)	Provide information about health consequences of physical inactivity (PA).
Behavior practice/rehearsal (8.1)	Prompt practice or rehearsal of the PA one or more times in a context or at a time when the performance may not be necessary, in order to increase habit and skill.

Definitions derived from Michie et al. [21]. PA = physical activity. BCT = behavior change technique.

**Table 2 ijerph-16-00331-t002:** Optional behavior change techniques named and numbered according to Michie et al. [21].

Goal setting (outcome) (1.3) Discrepancy between current behavior and goal (1.6) Review outcome goal(s) (1.7) Behavioral contract (1.8) Time management (according to the ‘Coventry, Aberdeen & London – Refined’ (CALO-RE) Taxonomy [48]) Self-monitoring of outcome(s) of behavior (2.4) Feedback on outcome(s) of behavior (2.7) Provide information on consequences of behavior to the individual (5.2 and 5.6 summarized) Provide info on where and when (According the CALO-RE Taxonomy [48]) Demonstration of the behavior (6.1) Prompts/cues (7.1) Habit formation (8.3)	Habit reversal (8.4) Generalization of target behavior (8.6) Graded tasks (8.7) Pros and cons (9.2) Reward and threat (summarizing 10.4, 10.5, 10.7, 10.9) Restructuring the physical environment (12.1) Restructuring the social environment (12.2) Avoidance/reducing exposure to cues for the behavior (12.3) Adding objects to the environment (12.5) Framing/reframing (13.2) Mental rehearsal of successful performance (15.2) Focus on past success (15.3) Self-talk (15.4)

**Table 3 ijerph-16-00331-t003:** Example for text messages.

Topic	Example
Knowledge transfer	Dear Ruth, did you know that regular endurance exercise helps boost your immune system?
Feedback on performance	Hello Mr. Meier, based on your online entries, I have seen that you had an active weekend. Gardening and a long walk on Sunday, congratulations!
BCT	Dear Katy, take your time to think about what prevents you from using the bike to go to work. Let’s discuss these obstacles next time☺. Have a good day.
Reminder	Good morning Ms. Bianchi, today is your first after-work swimming session. Don’t forget to pack your swimwear and have fun! Regards

**Table 4 ijerph-16-00331-t004:** Primary and secondary outcomes assessed in the Movingcall study.

Outcome Measure	Data Collection Instrument	Measure Point (Months)
Inclusion criteria	Adapted version of the Simple Physical Activity Questionnaire (SIMPAQ) [40]	0
Readiness for PA	Physical Activity Readiness Questionnaire (PAR-Q) [41]	0
Socio-demographic data	Commonly used items	0
Physical activity level	SIMPAQ [40]	0, 6, 12
	ActiGraph data of 7 days [55]	0, 6, 12
Variables used for tailoring	Self-compiled questionnaire on personal situation	0
	Self-compiled questionnaire on health restrictions, experience in PA and goals	0
Psychosocial determinants	Intention [56,57]	0, 6, 12
	Action planning [58]	0, 6, 12
	Outcome expectations [59]	0, 6, 12
	Self-efficacy [60].	0, 6, 12
	Perceived barriers [46]	0, 6, 12
	Coping strategies [61]	0, 6, 12
	Social support [62]	0, 6, 12
	Self-concordance [63].	0, 6, 12
	Knowledge about health enhancing PA and fitness rating [64,65,66]	0, 6, 12
Perceived stress-related exhaustion symptoms	Shirom–Melamed Burnout Measure (SMBM) [67]	0, 6, 12
Health-related quality of life	Short-Form 12 Questionnaire (SF-12) [68,69]	0, 6, 12
Feedback on aspects of the	Self-compiled questionnaire on satisfaction and homepage usability	3
Intervention	Self-compiled questionnaire on intervention depending on study arm	3, 6

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
