# Peer review of "Coaching and Prompting for Remote Physical Activity Promotion: Study Protocol of a Three-Arm Randomized Controlled Trial (Movingcall)"

_ijerph, 2019, doi:10.3390/ijerph16030331_

Round 1

Reviewer 1 Report

The abstract is missing information and is therefor unclear.

Line 15- you state that PA is a public health concern, but where? In the US, another county, worldwide?  Be specific.

Lines 20-21- What do you mean by 228 initially adults?  One cannot go from being an adult to not being an adult.  Or are you saying you started with that many subjects?  Change wording for clarity.

Line 22- change “lasting” to “long”

Line 22-23- you sated the N for the other two group but how many in the minimal intervention group?

Lines 25-27- Never actual state overall findings

Good overview of various existing methodologies in introduction

Line 38- change “does” to “do”

Reword lines 52-53

Lines 79-80- Do you mean various times throughout the day, or just varying times throughout a week, month, etc.?

As a trained Wellness Coach myself, I find the strategies in the protocol promising.  You have described your methodologies well.  I just have a few comments/questions.

Line 120- type “tree”

Clearly written figure.  Easy to follow.

Activity-profile is explained well

The phone coaching is very similar to the WellCoaches technique which is internationally recognized

Move Table 4 to line 338, where you first mention it in the text

Line 457- fix “

On line 457-458 you state that the option to analyze the data is not available for the wrist protocol.  How is this a valid analysis?

There are some grammatical errors in this paper.  I have identified some but you should take the time to review and edit the entire paper before submitting again.  There should also be commas places in several places where there are not.

It was stated several times that an intervention method, beyond a PA regimen, is needed to promote sustained PA.  Since these findings are so strong, is it ethics to expose a group to this when the participants are reaching out for help to change their PA behavior?

It also seems as those who do not have access to a computer or mobile device would be eliminated from this study.  Is there a way to provide the technology to this population so the study can be more inclusive?

Author Response

Our point-to-point response can be found in the pdf. Thank you and kind regards

Xenia Fischer

Reviewer 2 Report

The numerous writing errors (grammar, punctuation, missing words) detract from the potential value of the manuscript, but these issues can be easily remedied.

Additional suggestions/questions/comments are on the attachment.

Author Response

(The authors gave the same response as above.)
